# Impact of Force Scaling on Physician Fatigue in a Bilateral Tele-Ultrasound System

**DOI:** 10.3390/s25185894

**Published:** 2025-09-20

**Authors:** Eleonora Storto, Valerio Novelli, Antonio Frisoli, Francesco Porcini

**Affiliations:** IIM Institute, Scuola Superiore Sant’Anna of Pisa Italy, Via Alamanni 13b, 56010 Ghezzano, PI, Italy; eleonora.storto@santannapisa.it (E.S.); valerio.novelli@santannapisa.it (V.N.); antonio.frisoli@santannapisa.it (A.F.)

**Keywords:** tele-medicine, tele-ultrasound, EMG, muscle fatigue

## Abstract

Tele-ultrasound systems enable remote diagnostic imaging by transmitting both motion commands and haptic feedback between a sonographer and a robotic probe. While these systems aim to replicate conventional ultrasound procedures, they rarely address the physical strain typically experienced by sonographers. In this study, the effect of applying a force scaling strategy to haptic feedback on reducing muscular fatigue and task-induced stress during a realistic tele-ultrasound task is studied. First, a thorough operational and biomechanical analysis of the abdominal US procedure was performed to reconstruct a representative task in the laboratory. Then, a bilateral position–force tele-ultrasound architecture was implemented, and a total of 11 subjects performed the reconstructed remote ultrasound task under two randomized conditions: with and without force scaling. Surface electromyography (sEMG) signals were acquired from seven upper-limb muscles (posterior deltoid, trapezius, anterior deltoid, biceps, triceps, wrist flexors, and wrist extensors). Teleoperation-related stress was also assessed using a seven-item Likert-scale self-report questionnaire administered after each trial. Statistical significance was tested using Repeated Measures ANOVA for EMG data and the Wilcoxon signed-rank test for stress scores. The results showed a statistically significant reduction in muscle activation in 5 out of 7 muscles, and a clear mitigation of fatigue progression over time in the scaled condition. Additionally, perceived stress levels were significantly lower in the presence of force scaling in overall stress scores. These findings support the effectiveness of force scaling as a tool to enhance ergonomics in tele-ultrasound procedures without compromising the operator’s ability to perform the task. The proposed methodology proved robust and generalizable, offering valuable insight into the integration of human-centered design in tele-operated diagnostic systems.

## 1. Introduction

Tele-ultrasound is a promising technology in the field of tele-medicine that enables the remote execution of ultrasound (US) examinations. These systems help extend access to diagnostic services for routine screening and in emergency scenarios, even in remote or underserved areas, thereby contributing to the reduction in healthcare disparities.

This technology is a specific use case of the more general field of teleoperation. The first aim of teleoperation is to remotely replicate specific tasks successfully, providing the user with an embodiment feeling as immersive as possible. Thus, tele-ultrasound systems aim to enable physicians to remotely replicate US procedures thanks to robotic leader and follower devices. Typically, teleoperators are made by a user who directly interacts with a leader local robot, which sends a movement reference over a communication channel. This reference is realized by a follower robot, which interacts with a remote environment. The result of this interaction is measured through sensors and sent back to the user as feedback, realizing a bilateral teleoperation architecture. The nature of the reference and the feedback data determines the implemented teleoperation architecture [1]. In the tele-ultrasound use case shown in Figure 1, the user is a physician and the remote environment is the remote patient’s body. The follower robot end-effector mounts a US probe that interacts with the remote patient’s body according to the leader’s reference. Thus, both US images and force contact information are sent back to the user, enabling the possibility of force feedback. In this way, the physician, in addition to viewing real-time US images, receives force feedback that replicates the forces exerted on the patient’s body, making the procedure more natural, realistic, and precise (in terms of accurate positioning of the probe) despite the distance between leader and follower devices [2,3,4].

In traditional (local) US examinations, sonographers often suffer from work-related musculoskeletal disorders (WRMSDs) [5]. Those issues represent a significant concern for sonographers, with an incidence that, according to some studies, may reach up to 90% [5,6,7]. These disorders predominantly involve the upper body, especially in the neck, shoulders, lower back, wrist, and hand [5,6,7,8,9], compromising the physical well-being and occupational quality of life of these professionals. Going into more detail, these disorders arise from multiple factors. Among them, the need to maintain prolonged static postures [6,8,10,11,12], such as arm abduction during abdominal or thoracic scanning, leads to sustained muscle contractions that reduce blood flow, causing fatigue and pain [13]. Additionally, repetitive and precise movements of the hand and wrist, required to manipulate the probe, can result in cumulative microtraumas to tendons and nerves, increasing the risk of tendinitis and nerve compression syndromes [10,11,12]. Uncomfortable postures, often imposed by the patient’s position or suboptimal equipment configuration, tend to overload specific muscle groups [13], and the application of excessive force with the probe, especially when imaging obese patients to obtain quality images, further increases muscle load and injury risk [13,14]. It is also important to consider that the forces applied to the patient’s body depend both on the type of examination and the patient’s body mass index (BMI). The anatomical region where the greatest forces are recorded is the abdomen, where forces can reach up to 27.3 N [15]. These issues are aggravated by a workload that is often considerable. Although the frequency of US examinations varies significantly depending on the work setting, it is estimated that, on average, a sonographer performs between 10 and 20 US exams per day, during an 8 h shift [16,17]. The impact of these factors is compromising in the long term and difficult to mitigate, given that US procedures are performed by physicians themselves without the aid of support devices.

Even if tele-ultrasound systems show high potential in enabling the execution of a US task remotely, current solutions do not fully leverage robotics to improve physicians’ working conditions. In fact, tele-ultrasound systems mainly focus on replicating the procedure, without adequately addressing these ergonomic issues [18,19,20]. This results in a missed opportunity to reduce operator fatigue and muscular effort, improving the safety and sustainability of US practice. Moreover, the WRMSDs’ issues are expected to get worse during a teleoperation. It is well established in the state of the art that teleoperation tasks increase mental workload and fatigue on the operator with respect to executing the same task locally [21,22,23,24]. The reason besides this lies in the fact that, even if teleoperation aims to perfectly recreate human skills remotely, the usage of robotic devices makes the task always less natural during execution, raising the workload. This aspect is studied in the teleoperation field, and the solution may be task-dependent or rely on shared autonomy [25]. It is important to reduce the impact of WRMSDs on physicians. In particular, in diagnostic applications, it is not acceptable that the quality of the service provided to patients is subject to a decline due to working conditions.

Considering the aspects described, the scientific literature shows that tele-ultrasound investigates and develops methods to address WRMSDs, for example, through the implementation of force scaling in the feedback delivered to the operator. The use of a force scaling factor has been implemented, for instance, in the works of Santos et al. [3], Si et al. [26], and Storto et al. [27]. In the first two cases, however, its use is not specifically aimed at reducing the force rendered to the physician for the purpose of alleviating WRMSDs, nor is this effect analyzed. Conversely, the work by Storto et al. [27] aimed to evaluate the performance of a specific bilateral tele-ultrasound architecture under realistic working conditions, including the presence of time delays, the application of US-required contact forces, and the possibility to use a scaling factor on the force feedback to address operator fatigue.

Although the scientific literature presents tele-ultrasound approaches that aim to address WRMSDs (even if not as the main claim), such as the use of force scaling in feedback mechanisms, there is, to the best of our knowledge, no study that directly and objectively assesses operator fatigue during the execution of tele-ultrasound examinations.

The motivation for this study arises from this gap. The opportunity offered by robotic devices in tele-ultrasound tasks lies in the ability to provide force feedback by choosing to sacrifice transparency (that is, the fidelity with which interaction forces with the patient are transmitted to the physician without distortion), thereby delegating the physical effort and fatigue to the robotic manipulator rather than to the physician. Leveraging the system’s robotic devices to alleviate fatigue during task execution can also have a direct effect on perceived stress. Thus, the introduction of appropriate scaling factors is potentially able to both reduce WRMSDs and the stress during a tele-ultrasound task.

The aim of this study is to evaluate the effectiveness of scaling the force rendered to the physician using a bilateral tele-ultrasound system in reducing muscle fatigue, as measured through muscular activity recorded by an EMG, and in reducing the teleoperation task-related stress, evaluated by a self-report questionnaire based on a standardized Likert-scale. The evaluation focuses on whether the scaling factor can effectively reduce physical fatigue, diminishing the risk of developing WRMSDs and the teleoperation task-related stress, without compromising the execution of the US task or the accuracy of the diagnosis. An abdominal US procedure was chosen as the reference task. Thus, the reference procedure was analyzed, and a biomechanical analysis of a sonographer performing such a task was carried out. The procedure was reconstructed in a laboratory through a tele-ultrasound position–force measured (PFm) architecture to be as realistic as possible: the task foresees the regulation of an average contact force during the procedure on an abdominal phantom, and the operator’s posture and timing closely replicate those of a conventional, in-person US examination. To evaluate the effect of a force feedback scaling factor on fatigue and teleoperation-related stress while performing the remote abdominal US procedure, 11 healthy volunteer subjects were involved to behave as sonographers performing the realistically reconstructed task both in scaled and non-scaled conditions. For each subject, EMG signals of seven muscles were acquired to evaluate fatigue, while a self-report questionnaire based on a standardized Likert-scale was filled out to evaluate stress. The results obtained by realistically replicating the task have demonstrated with statistical relevance that the introduction of an appropriate scaling factor leads to a strong reduction in fatigue for 5 out of 7 muscle groups and a clear decrease in stress.

This paper is organized as follows. Section 2 considers a standard abdominal US procedure as the reference task and provides an in-depth operational and biomechanical analysis of the procedure. In Section 3, the experimental protocol, including the description of the custom tele-ultrasound system and of the EMG signal acquisition device, for evaluating the effect of force scaling on sonographers’ fatigue is described. Section 4 explains the evaluation metrics and data analysis, while Section 5 reports the achieved results in terms of reducing EMG activation level and leveraging the task-related stress. Lastly, Section 6 concludes with a discussion of the obtained results and highlights future directions.

## 2. Material and Methods

The abdominal region was selected as the target anatomical area due to its known biomechanical force-related (then fatigue) challenges in US procedures due to factors like air, fecal matter, and adipose tissue in the abdominal area [28]. In order to recreate a realistic abdominal US procedure in the laboratory, operational aspects, such as duration of contact and rest phases, and applied forces, should be investigated and taken into account. Among these aspects, a biomechanical analysis of a sonographer while performing such a procedure has been carried out to identify the pose, qualitative kinematics, and involved muscles. The importance of that data is straightforward, considering that a reduction in WRMSDs may be easily related to a reduction in the activation of the EMG levels.

### 2.1. Operational Aspects of Abdominal US Procedures

Real abdominal US procedures engage sonographers in repetitive and physically demanding tasks, requiring sustained arm postures and controlled application of force. Thus, it is important to correctly identify both the typical duration and the usual applied forces in this procedure.

The total duration of each US examination includes image acquisition, patient reception, report writing, and equipment preparation. Although no precise standard times exist, and the duration may vary, some abdominal US studies report the following time intervals. In the study by Reuss et al. [29], focused on upper abdomen and kidney US examinations, the duration of the US scan itself was reported as 12.3 min, while the total procedure time, including patient preparation and documentation, was 21.3 min. In the work by Teichgräber et al. [30], a minimum total time of 24 min was reported to complete an abdominal US examination, of which 6 min were dedicated to the actual US scanning. On the other hand, Village et al. [8], in their study, report effective scanning times of 12 and 8 min, without specifying the total duration of the examination.

The forces applied to the patient vary based on abdominal composition and BMI, as highlighted in the study by Dhyani et al. [31]. The average force ranges from 7.5 N for patients with a normal BMI (18.5–25) to 9.8 N for a high body mass index (BMI > 25), with an estimated average of 7 N. Maximum forces are reported as 17.3 N (normal BMI), 22.4 N (high BMI), and 27.3 N (general patients).

These durations and the exchanged force values during abdominal US examinations serve as reference benchmarks to ensure the biomechanical and procedural realism of the task reconstructed in the laboratory.

### 2.2. Biomechanical Analysis

The biomechanical analysis of the task has been conducted through visual observation, EMG recording, and in accordance with the literature. The result of this analysis leads to defining the pose assumed during the task, the qualitative movement, and the involved muscles.

During a typical abdominal US procedure (see Figure 2), the sonographer sits next to the patient facing the screen, with the arm extended toward the target anatomical region. The starting posture was defined based on visual observation of sonographers and the literature data. Referring to Figure 3, a sonographer typically sits with the elbow flexed at 90° and shoulder abduction α greater than 30° in the frontal plane [11,12]. To the best of the authors’ knowledge, there is no information about the forearm external rotation β value during the task. However, based on visual observation of the task, the forearm should approach the patient’s body in as comfortable a configuration as possible for the sonographer, allowing the forearm to be positioned in a high manipulability area of the human workspace [32,33].

Thus, the identified configuration can be assumed as the starting pose for the task and is visible on the left of Figure 3. It should be noted that the rotation of the wrist was not taken into account due to highly variable hand orientation dependence. This orientation is influenced by probe handling, and thus, by probe type and sonographers’ preferences.

From the starting pose, the sonographer lowers their hand to contact the patient’s body and raises it when the procedure ends. The probe orientation may change during contact, but this was neglected due to minimal impact on applied pressure (thus on the force-caused WRMSDs).

Even if the sonographer typically changes the orientation of the probe according to the images, this does not significantly change the applied pressure (that is, the cause for WRMSDs). Then, a fixed orientation was assumed. The qualitative movement of this simplified abdominal US procedure is modeled as a straight line perpendicular to the transversal plane path following.

The selection of monitored muscles was guided by their functional role in supporting the movements and postures required during an abdominal US procedure. The anterior deltoid is primarily involved in shoulder flexion (lifting the arm forward and upward) and abduction (moving the arm away from the body), providing the main driving force for reaching the target anatomical region. The posterior deltoid acts as a stabilizer and counterbalance to the anterior deltoid during these movements, and contributes to shoulder extension when the arm is lowered to rest. The trapezius supports the abduction movement by stabilizing the scapula [34]. The biceps contribute to forearm flexion and supination, whereas the triceps act as an antagonist, extending the elbow and providing stabilization during the task [35]. Finally, the wrist flexors and extensors play a crucial role in maintaining a firm and controlled grip on the probe, enabling precise application of contact force and fine manipulation, while also stabilizing the wrist joint [36]. The muscles involved in this task are illustrated in Figure 3.

### 2.3. Abdominal US Procedure Laboratory Reconstruction

To experimentally evaluate the effectiveness of force feedback scaling during abdominal US tasks, it is essential to reconstruct the procedure in a laboratory setting with a high degree of realism, reproducing the standard working conditions of a medical sonographer. The aim is to replicate the procedure with sufficient realism to recreate comparable fatigue conditions on the target muscles with respect to the real task. This reconstruction should faithfully replicate both the operational aspects and the biomechanics involved in performing a conventional abdominal US examination, including the operator’s posture, movement dynamics, contact duration, and force application.

To enable a focused analysis of fatigue while maintaining experimental control, a simplified version of the task was implemented. Specifically, the physician’s initial posture replicated the one identified in the biomechanical analysis, characterized by a shoulder abduction greater than 30°, elbow flexion at 90°, and forearm rotation exceeding 45°, placing the limb in a region of high manipulability within the human workspace. From this starting pose, the task foresees a single, consistent motion: lowering the hand vertically to bring the probe into contact with the phantom and lifting it again at the end of the contact phase. More complex actions, such as probe sliding, wrist rotation, or reorientation for image optimization, were deliberately excluded from this protocol. These gestures were neglected because they typically involve minimal force exchange and are highly variable depending on probe type and user preference (in the case of the orientation of the hand). Moreover, image quality (which in real scenarios is often adjusted through subtle reorientations) can be assumed to be directly proportional to the applied contact force. This allowed for a reliable simplification: by focusing on controlled force modulation alone, the task could retain physiological relevance while remaining experimentally tractable.

As for the task timing, a realistic clinical scenario was considered. The worst-case literature reports that a typical 8 h workday may involve up to 22 abdominal US exams, each averaging around 22 min in total duration, with approximately 12 min of active scanning and 10 min allocated to preparation and reporting. In order to replicate this distribution of workload and rest without requiring excessive time from participants, the experimental task was temporally compressed. Each simulated exam was scaled down to a 22 s trial, with 12 s of sustained probe contact followed by 10 s of rest. This scaling preserved both the ratio between exertion and recovery phases and the total number of repetitions (22 cycles), thereby ensuring that the pattern of muscle loading reflected realistic working conditions while remaining feasible for a lab setting.

Regarding contact force, the reconstructed task adopted a target force range between 6 and 12 N, based on values reported for abdominal ultrasound procedures. The task foresees maintaining a steady force within this range during each 12 s contact phase. This constraint ensured the biomechanical realism of the task while offering a simplified yet effective surrogate for maintaining adequate image quality. In fact, image quality directly depends on the applied force [37]. The participant was instructed to maintain the applied force within the specified range, supported by a visual feedback displayed on a screen. The screen showed the real-time force applied on the slave device, along with two horizontal reference lines representing the lower and upper bounds of 6 N and 12 N, respectively. This created a target force band within which the participant aimed to keep the contact force with the phantom.

The muscles included in the analysis are those reported in Section 2.2, namely the posterior deltoid, trapezius, anterior deltoid, biceps, triceps, and the wrist flexors and extensors.

## 3. Experimental Protocol

The purpose of the experimental study is to assess the effectiveness of a force scaling strategy in reducing muscle fatigue experienced by sonographers during abdominal US examinations performed via a tele-ultrasound robotic system. In order to ensure that the task remains physiologically and operationally comparable to a conventional clinical scenario, the tele-operated procedure must replicate real US practice as closely as possible. To this end, the laboratory setup described in Section 2.1 was adopted as the operational reference for all subjects involved in the experimental trials.

The experimental setup was designed to closely mimic the operational and biomechanical aspects of a conventional in-person US examination, as described in the previous sections. In particular, participants are asked to reproduce the posture, timing, and force application typically observed during routine abdominal scans. The only variable introduced in the experimental protocol is the scaling factor in the tele-ultrasound system itself, which can be configured either to provide or not provide the force scaling. This allows for a direct comparison between the two conditions (scaled vs. non-scaled) under otherwise identical task execution parameters.

### 3.1. Experimental Setup

The experimental task, derived from the reconstructed abdominal US procedure, was implemented using a custom-developed tele-ultrasound system. In parallel, surface electromyography (sEMG) was employed to objectively monitor muscle activation throughout the task. In the following, the hardware and configuration of both the robotic devices and the acquisition system are described, along with a description of the experiment.

#### 3.1.1. Tele-Ultrasound System

The tele-ultrasound system, visible in Figure 4, is made by a leader robotic device (on the left), controlled by the user, and a follower robot arm (on the right) that interacts through the US probe with the remote phantom. The devices exchange data over a communication channel. In particular, the leader sends the velocity reference (as information of the pose assumed by the sonographer) and the follower sends back the interaction force.

The leader robotic device (Figure 4a) is an impedance-based parallel kinematics haptic interface. This robot allows for three translational degrees of freedom (DoFs) and a maximum of 15 N of rendered force. Each motor joint is sensorized through an encoder, and the cable-driven transmission guarantees a 1:10 reduction ratio. To allow complete pose control, the interface has been equipped with a custom spherical joint sensorized with three encoders. A fictive probe, modeled on the basis of the real US probe, has been assembled at the end-effector to give the user the sensation of controlling the real probe. The interface is controlled in a force open-loop, while gravity and friction are compensated in feedforward through look-up tables.

The follower robotic arm (Figure 4b) is the commercialized collaborative robot Doosan M0609 (Doosan Robotics, Gyeonggi-do, Republic of Korea). The arm performs six joints, allowing for complete end-effector pose control. The low-level control (LLC) is a proportional-integrative controller with gravity compensation (PI-G) that follows the leader’s velocity reference. To guarantee a good force feedback, the M0609 end-effector has been equipped with an ATI Gamma 6-axis force/torque sensor, and a real US probe has been attached to the sensor as an interacting tool with the phantom. The sensor has been calibrated to reject the inertia of the attached probe, guaranteeing that no force is sent back to the user in free motion.

The complete teleoperation architecture has been represented as a block diagram in Figure 5. The user is represented as a force $F_{H}$ applied to the leader $G_{L}$ along with the feedforward compensations $F_{CL}$ and the feedback force $F_{LD}$, which is the force the leader should display to the user. The leader velocity $v_{L}$ is sent through the communication channel as a reference to the follower $G_{F}$, and the local low-level $PI_{F}$ control loop with gravity feedforward compensation $F_{GF}$ is closed over the actual $v_{F}$ follower velocity. The interaction force $F_{E}$ with the environment (which is the abdominal phantom in this use case) is recorded by the sensor $H_{F}$ and sent as feedback to the user. The adopted architecture is a typical position–force measured teleoperation architecture [1]. For the purpose of this work, no delay was considered introduced by the communication channel; thus, no passivity layer was implemented to stabilize the architecture [38]. Although latency is a natural component of remote systems, it compromises the stability of force feedback teleoperation. While passivity layers can be employed to mitigate this issue, they introduce additional damping and alter the system’s dynamics. More importantly, delay-induced instability and the force scaling strategy under investigation may mutually influence each other: instability can exacerbate muscular fatigue and stress, while force scaling (designed to reduce them) may simultaneously enhance system stability. To prevent this mutual influence and ensure a rigorous evaluation of the scaling effect, the two phenomena were intentionally decoupled in the present study, sacrificing the realistic presence of time delay in the communication channel. Nonetheless, the presence of time delays in tele-medicine applications is unavoidable. This represents a limitation of the present study due to the need to evaluate muscle fatigue only in relation to the scaling factor.

To reduce operator fatigue during prolonged tele-ultrasound procedures, a force scaling factor μ was introduced on the feedback channel, which is visible in Figure 5. In this study, the value of μ was fixed at 0.5. While this is not a universally optimal value for all potential users of the system, it was kept constant to remove a variable component from the experimental setup, ensuring that force evaluation was not influenced by additional uncontrolled factors. The choice of μ = 0.5 follows our previous work [27], in which the same value was selected by the user after a familiarization phase. This factor attenuates the amplitude of the measured interaction force before rendering it to the user. While this approach inevitably reduces system transparency, it is expected to significantly alleviate muscular effort. Importantly, in teleoperation, the most transparent force feedback is not always the most effective for task performance: as discussed in [39], the goal is not to maximize transparency per se, but to optimize feedback to support successful task execution. In the specific context of tele-ultrasound, the primary feedback modality is the US image itself, which (assuming sufficient resolution and real-time quality) provides direct and immediate information about task success. The force feedback, therefore, assumes a complementary role, supporting the perception of contact quality and force control. When properly scaled, haptic feedback can convey more intuitive and informative cues than traditional numerical force indicators shown on a display, which are commonly used in conventional tele-ultrasound systems [40,41]. Nonetheless, the haptic channel remains essential to ensure a natural, realistic, and intuitive user experience, as emphasized in [42].

To complete the architecture, the US system used in this study is the Esaote MyLab Sigma (Esaote S.p.A., Genoa, GE, Italy), equipped with a convex-array transducer (model AC2541) mounted on the robotic follower arm. This probe operates over a frequency range of 1.0 MHz to 8.0 MHz and is suitable for various clinical applications, including abdominal, obstetric, gynecological, and vascular imaging. The AC2541 probe is compatible with multiple US platforms within the Esaote MyLab family, such as the MyLab Six, Seven, Alpha, and Gamma systems, ensuring high versatility and clinical reliability. The probe was mechanically secured to the end-effector of the follower robot and acoustically coupled to a phantom surface using standard US gel. While the MyLab Sigma maintained full diagnostic image quality, and real-time video output was streamed to the operator side, image interpretation was not required during the task. Consequently, the operator’s role was limited to positioning and maintaining appropriate contact force, which served as a controlled surrogate for effective image acquisition. All other functionalities of the US system (such as control of imaging parameters and full image display) were available and remotely replicated, although they were not actively used during the experiment. This approach was adopted because the magnitude of the force with which the clinician interacts with the patient’s body is directly related to the acquisition of ultrasound images [37]. From an early study of the average forces required for performing an abdominal ultrasound, which were replicated by the subject in a controlled manner, it was considered unnecessary to involve ultrasound image visualization for the purpose of fatigue assessment. Furthermore, given the provision of continuous real-time visual feedback of the applied contact force, the inclusion of concurrent ultrasound image visualization was deliberately excluded to prevent cognitive interference. This approach ensured that the participant’s attentional resources were fully dedicated to maintaining force within the prescribed range, a critical factor for the valid assessment of operator fatigue.

#### 3.1.2. EMG Signal Device Description

A wearable surface electromyography (sEMG) signal acquisition device was used to determine muscle fatigue in each subject. These signals represent the electrical potentials sent by muscle fibers and are used to assess muscle contraction and fatigue.

Each subject was equipped with a compact, wearable device capable of acquiring up to eight channels of sEMG signals [43]. The device was mounted on the upper arm to avoid interference with electrode placement. The device features an STM32F407VG microcontroller (STMicroelectronics NV, Geneva, Switzerland) (ARM Cortex-M4 @ 168 MHz) with 1 MB of Flash memory and 192 kB of RAM. Signal acquisition is performed using a 24-bit Texas Instruments ADS1298 (Texas Instruments, Dallas, TX, USA) analog-to-digital converter, capable of sampling rates up to 16 kHz. Wi-Fi communication is handled by a Murata SN8205 module, interfaced via UART at 921,600 bps and supporting TCP/IP transmission. The software architecture is divided into three main components: firmware for the main CPU, firmware for the Wi-Fi module, and a PC-based client for data acquisition and visualization.

Each subject was equipped with the sEMG signal acquisition device, consisting of a band placed on the upper arm (to avoid interfering with electrode placement). This band housed the battery and the signal acquisition board. The sEMG signals were transmitted via Wi-Fi over a local network to the master local computer using the TCP/IP communication protocol. All sEMG signals, along with the leader and follower data, were managed within the same logging system and synchronized at a common sampling frequency of 1 kHz.

The electrodes were connected to this board and then positioned on the subject’s arm in the regions corresponding to the muscles under investigation. Specifically, seven electrodes were used to acquire sEMG signals, positioned on the trapezius, posterior deltoid, anterior deltoid, biceps, triceps, wrist flexor, and wrist extensor muscles. Additionally, two electrodes were used for the ground and placed on the epicondyles. The electrode placement is illustrated in Figure 6.

In addition to the sEMG signal acquisition during each trial of the experimental procedure, the device was also used to collect the Maximum Voluntary Contraction (MVC) values for each muscle under investigation, for each subject individually. This allowed for signal normalization and comparison across participants. The MVC signals were recorded during a preliminary phase, conducted prior to the execution of the experimental trials. In this session, participants were instructed to contract each target muscle separately and with maximum effort, following a standardized sequence. Each contraction was supervised to ensure correct muscle engagement and to minimize cross-talk or compensatory movements.

The electrodes were positioned according to SENIAM guidelines (http://www.seniam.org/, accessed on 10 April 2025), ensuring standardized and effective measurement of muscle activity across the different subjects who participated in this study.

### 3.2. Task Design and Participants

Experimental trials were conducted on 11 healthy adult volunteers, who were instructed to replicate the US task as reconstructed in the previous sections. Rather than involving trained sonographers, this study was designed around participants with no prior US experience. This decision was methodologically appropriate because the task, as defined in Section 2.3 and without distorting results, involved a simple, repeated compressive gesture that could be performed reliably without requiring diagnostic skills or image interpretation. In fact, the task involved executing a series of compressive interactions using a tele-ultrasound system, following the motion and posture detailed in Section 2.3. To ensure that subjects without prior sonographic experience could perform the task accurately, image quality was indirectly controlled with the applied contact force, which had to be maintained within a 6–12 N target band. The screen showed the real-time force applied on the follower device, along with two horizontal reference lines representing the lower and upper bounds (of 6 N and 12 N, respectively). This created a target force band within which the participant aimed to keep the contact force with the phantom. This simplification allowed even untrained participants to execute a physiologically meaningful gesture consistent with real clinical conditions. Moreover, the absence of task-specific muscle memory (which experienced sonographers typically develop) made signs of muscular fatigue more evident and measurable. As for stress, it was deemed relevant to involve subjects unfamiliar with teleoperation tasks to enhance sensitivity in detecting cognitive workload and task-induced pressure.

Each participant wore surface EMG electrodes placed on the muscle groups identified in previous analyses (see Section 2.2 and Section 2.3), and was fitted with an EMG acquisition device described in Section 3.1.2. A baseline Maximum Voluntary Contraction (MVC) protocol was executed for each monitored muscle prior to the task. Trials were performed under two experimental conditions: one with the force scaling factor active (scaled), and one without it (non-scaled). The order of the conditions was randomized across subjects to avoid sequence bias.

For each condition, the protocol included the following steps:Familiarization phase, during which the subject interacted freely with the teleoperation system, both in free motion and in contact, to become accustomed to its dynamics.Pre-task rest period, to ensure muscle recovery and a consistent starting state.Task execution (as reconstructed in Section 2.3), consisting of 22 cycles of controlled probe contact and release. Each cycle included 12 s of active contact (during which the subject maintained the displayed contact force within the prescribed 6–12 N range) followed by 10 s of rest. Timing was externally managed and verbally indicated by the experimenters.Post-task rest and questionnaire, allowing for muscular recovery (minimum 4 h before the next session), and administration of a stress perception self-report questionnaire detailed in Section 4.2.The same protocol was repeated identically under the other condition (scaled or non-scaled), followed by an identical recovery and questionnaire phase.

For each subject, EMG activity and stress questionnaire responses were collected separately for both experimental conditions, enabling a within-subject comparison of muscle fatigue and perceived stress under scaled and unscaled force feedback.

## 4. Evaluation Metrics and Data Analysis

This section outlines the methodologies adopted for analyzing the physiological and subjective data collected during the experimental trials. The goal is to evaluate muscle activity, fatigue, and perceived teleoperation-related stress under the two experimental conditions: scaled and non-scaled. The analysis is organized into three main parts. First, the full processing pipeline of the surface electromyography (sEMG) signals is described, including acquisition, preprocessing, segmentation, normalization, and feature extraction. This allows for the quantification of muscle activation and fatigue across different time windows related to contacts and muscle groups. Second, the subjective perception of stress is assessed using self-report questionnaires based on validated Likert-scale items adapted from established instruments. Finally, statistical analyses are performed to evaluate the effects of the two conditions, including normality checks and the application of Repeated Measures ANOVA to compare muscle activation trends and fatigue, as well as a Wilcoxon signed-rank test to assess differences in perceived stress between non-scaled and scaled trials.

### 4.1. EMG Signal Processing

In this chapter, the complete process of acquisition, preprocessing, normalization with respect to MVC, and RMS calculation of the sEMG signals collected during the experiment is described, with the aim of evaluating muscle activity and fatigue in the subjects involved under the two experimental conditions: non-scaled and scaled.

The following describes the data analysis process of the sEMG signals, carried out consistently for each subject.

Using the sEMG signal acquisition device described in Section 3.1.2, raw sEMG signals were recorded from seven muscles in the kinematic chain of the arm: the posterior deltoid, trapezius, anterior deltoid, biceps, triceps, wrist flexor, and wrist extensor. These muscles were selected in accordance with the state of the art, as described in Section 2.2. The data were collected both during the experimental trials and in the preliminary phase dedicated to estimating the Maximum Voluntary Contraction (MVC) for each muscle. The sEMG signals were transmitted via Wi-Fi over a local network using the TCP/IP communication protocol. The sampling frequency (Fs) of the EMG signals and of the entire master–slave system was the same and equal to 1000 Hz, thus ensuring synchronized data acquisition.

During each experimental trial, subjects performed 22 consecutive contacts, each lasting 15 s and separated by 11 s of rest. To analyze muscle activation specifically during the contraction phases, it was necessary to isolate the corresponding periods within each trial. For this purpose, the force signal exerted by the subject was used as a reference to identify the onset and duration of each contact. Specifically, for each subject and each trial, 22 windows were segmented from the force signal, based on the criterion of exceeding a 6 N threshold. These segments were then used to extract the corresponding portions of the sEMG signals, effectively isolating the muscle activity related to contact and excluding rest periods. This threshold-based approach served as an indicator of the onset of active interaction with the object.

Starting from the raw signals, a preprocessing procedure was carried out to make them interpretable and suitable for quantitative analysis. The filtering was performed by sequentially applying the following digital filters:A 50 Hz notch filter, to eliminate noise generated by the power line. sEMG signals, due to their low amplitude and broadband nature, are highly susceptible to power line hum, which is difficult to remove as it overlaps with the signal and lacks a distinctive waveform [44];A 5 Hz high-pass filter, to remove low-frequency components caused by slow movements, artifacts, or baseline noise;Full-wave rectification of the signal to convert all signal values to positive, facilitating the quantification of muscle activation regardless of signal polarity;A 15 Hz low-pass filter, used to extract the envelope of the signal, i.e., the smoothed profile of muscle activity intensity over time.

The last three points are recommended in the ISEK standards for EMG signal acquisition [45]. It is important to note that this same preprocessing procedure was applied both to the signals acquired during the experimental trials and to those recorded during the preliminary phase dedicated to the estimation of MVC values.

During a preliminary phase, each participant was asked to perform isolated and sequential maximal voluntary contractions (MVCs) for each muscle. From these recordings, the peak RMS value of the sEMG signal was extracted and used as the reference MVC value for that specific muscle and subject. Subsequently, the sEMG signals recorded during the experimental trials were normalized by dividing each data point by the respective MVC value. The resulting normalized signal, expressed as a ratio between 0 and 1, reflects the relative level of muscle activation with respect to the individual’s maximal voluntary contraction [45]. This normalization procedure enables meaningful comparisons across subjects and trials and allows for a more consistent interpretation of muscle activation and fatigue patterns.

As introduced in Section 3.1.2, the analysis of the sEMG signal aims to characterize muscle contraction and fatigue. In order to evaluate muscle activation at each contact event and to compare performance between the non-scaled and scaled trials, the Root Mean Square (RMS) value of the sEMG signal was computed for each subject, muscle, and trial within the predefined contact windows. This approach enabled a direct quantification of activation intensity associated with specific phases of the task under different experimental conditions.

### 4.2. Perceived Stress Assessment

To assess participants’ subjective experience during tele-ultrasound task execution, all subjects completed a self-report questionnaire after each experimental condition (scaled and non-scaled). The instrument comprised seven closed-ended items designed to capture task-induced distress, and each item was rated on a 5-point Likert-scale (from 1 = not at all true to 5 = completely true). The items were adapted from well-established and validated instruments in the literature (see Table 1), including the Perceived Stress Scale (PSS) [46], the State–Trait Anxiety Inventory (STAI) [47], and the NASA-TLX [48], particularly focusing on the stress dimension.

The questionnaire was administered immediately after each trial session (scaled and non-scaled), using a paper form. For each condition, a total stress score was calculated as the sum of all item scores, ranging from 7 to 35 (where 7 indicates a low stress level and 35 a high stress level perceived during the task). The structure of the instrument allows for a focused assessment of perceived psychological stress linked to the execution of the teleoperation task, rather than general affective states. Internal consistency of the seven-item scale was assessed using Cronbach’s alpha, despite the small sample size and the exploratory nature of this pilot study. Nevertheless, all items were adapted from psychometrically validated instruments and retained their original semantic focus during the adaptation process.

The questionnaire provided a quantitative yet subjective index of teleoperation-induced distress, allowing for the evaluation of the impact of force feedback scaling not only on physical fatigue but also on the psychological load experienced by the operator.

### 4.3. Statistical Analysis

In this section, the statistical strategy adopted to analyze both the processed sEMG data and the self-reported questionnaire responses is presented. The objective is to evaluate potential differences in muscle activation and perceived task-related distress between the two experimental conditions: non-scaled and scaled.

A preliminary step in the statistical analysis of the acquired sEMG data was to verify the normality of the distribution of mean RMS values across subjects for each muscle and experimental condition (non-scaled vs. scaled). This assessment is fundamental to determining the appropriateness of applying subsequent parametric tests for comparing muscle activation between conditions. The Kolmogorov–Smirnov test was applied to these data after normalization (centering and scaling) to evaluate their compatibility with a standard normal distribution. The results showed no significant deviation from normality (p>0.05) for all muscles and conditions, confirming the validity of the normality assumption and supporting the use of parametric statistical methods for further analysis. As an additional robustness check, given the small sample size of this study, the Shapiro–Wilk test was also performed and similarly confirmed the normality of all distributions (p>0.05). Since the RMS values of muscle activation were found to follow a normal distribution, parametric statistical methods were appropriate to compare the two experimental conditions: non-scaled and scaled. Specifically, a Repeated Measures ANOVA (RM ANOVA) was conducted for each muscle to assess whether significant differences existed between the scaled and non-scaled trials. Given that only two conditions were compared, the sphericity assumption was inherently satisfied, so Mauchly’s test was not applied, and no corrections (e.g., Greenhouse–Geisser) were necessary. The RM ANOVA was selected because it accounts for the within-subject correlation of repeated measurements, thereby increasing the statistical power and reducing variability attributable to inter-subject differences. Given that multiple RM ANOVAs were performed once per muscle, adjustments for multiple comparisons were applied. Accordingly, both the Holm correction, applied to control the family-wise error rate, and the Benjamini–Hochberg (BH) correction, applied to control the false discovery rate, have been reported.

The statistical analysis to evaluate stress levels from the self-report questionnaire was conducted using the following methods. For each subject and condition between non-scaled and scaled, the total stress score was computed by summing the responses across all seven items separately. The Wilcoxon signed-rank test, a non-parametric test suitable for paired ordinal data, was then applied to compare stress levels between the non-scaled and scaled conditions, both for each individual question and for the overall total scores. As for the Wilcoxon signed-rank tests, corrections for multiple comparisons (Holm and Benjamini–Hochberg) were applied as described above.

## 5. Results

According to the rationale of this study, the experimental results revealed a clear relationship between the application of the force scaling factor and both muscular fatigue and perceived stress. The results related to the analysis of the sEMG signals are visible in Figure 7 and Figure 8, while Figure 9, Figure 10 and Figure 11 show the results related to the analysis of the self-report questionnaire. Those results were divided into two subsections for better readability.

### 5.1. Muscle Fatigue Result Analysis

The RMS values of muscle activation during the 22 contacts for each muscle are shown in Figure 7 in the two test configurations: the non-scaled configuration at the top, and the scaled one at the bottom. This figure proposes, as an example, experimental results achieved by subject 2 for the sake of space. The RMS values were normalized, as described in Section 4, and therefore range between 0 and 1. However, to improve the readability of the plots, a scale adapted to the actual range of the obtained values was used and kept consistent for the same muscle across both the non-scaled and scaled configurations (vertically in Figure 7). This choice was consistently applied to all the plots presented in this section. From Figure 7 it is evident that for all seven analyzed muscles, the RMS values of muscle activation are higher in the non-scaled condition and lower in the scaled condition. In the scaled configuration, muscle activation remains relatively constant over time, with no clear signs of fatigue progression during the trial. Conversely, in the non-scaled configuration, a progressive increase in RMS values is evident, indicating growing muscular fatigue as the number of contacts increases. The absence of the scaling factor leads to greater neuromuscular involvement over time, resulting in increased fatigue. In contrast, the application of the scaling factor reduces the intensity of the applied force, leading to lower neuromuscular demand and the absence of clear fatigue indicators. For completeness, the mean ± standard deviation of muscle activation (%MVC) for each muscle in the two experimental conditions is summarized in Table 2, allowing a direct quantitative comparison of the observed effects. These findings confirm that typical US levels of applied force can induce fatigue, whereas lower force levels, achieved through scaling and robotic aid, help mitigate this effect, making it easier to control muscular activation during task execution.

The findings reported for subject 2 were further confirmed by the statistical analysis carried out over all the involved subjects. In fact, Figure 8 shows the results of the statistical analysis performed using Repeated Measures ANOVA (RM ANOVA) on all subjects, as described in Section 4. For each muscle, two boxplots are presented representing the distribution of the normalized mean RMS values for the two experimental conditions: non-scaled (in red) and scaled (in blue). The graph displays asterisks indicating statistically significant differences between the two conditions: non-scaled and scaled. Corresponding unadjusted, Holm, and BH *p*-values, and partial eta squared ($\eta_{p}^{2}$) values for each muscle are reported in Table 2. Significance levels refer to p-values and are denoted as follows: *** for *p* < 0.001, ** for *p* < 0.01, * for *p* < 0.05, and no asterisk for ≥0.05. Partial eta squared values are provided as a measure of effect size, with conventional benchmarks of approximately 0.01, 0.06, and 0.14 representing small, medium, and large effects, respectively. Starting from the Holm correction, which controls the family-wise error rate and represents the primary comparison, as it is more conservative than the BH correction, it can be observed from the results that a statistically significant difference between the two conditions was found for the biceps (RM ANOVA, $p_{\mathrm{Holm}}=0.0469$), wrist flexors (RM ANOVA, $p_{\mathrm{Holm}}=0.0430$), and wrist extensors (RM ANOVA, $p_{\mathrm{Holm}}=0.0453$). This effect can be attributed to a marked reduction in the need to maintain a strong grip (and therefore a high impedance of the entire arm) on the fictive probe during the task, due to the decreased feedback force perceived by the subject thanks to the scaling factor. However, no statistically significant differences were observed for the posterior deltoid (RM ANOVA, $p_{\mathrm{Holm}}=0.1322$), trapezius (RM ANOVA, $p_{\mathrm{Holm}}=0.1338$), anterior deltoid (RM ANOVA, $p_{\mathrm{Holm}}=0.0923$), and triceps (RM ANOVA, $p_{\mathrm{Holm}}=0.1033$). This finding suggests that the shoulder muscles play a fundamental role in bearing the entire arm load during task execution, independently of the movement, and are therefore not directly involved in the movement under consideration but serve a supporting role. Similarly, the triceps primarily contributes to stabilizing the arm during extension and grip force control, indirectly supporting the muscles more actively involved in hand and wrist movement. Considering the Benjamini–Hochberg (BH) correction, the results change slightly. Under this less conservative procedure, both the anterior deltoid (RM ANOVA, $p_{\mathrm{BH}}=0.0431$) and the triceps (RM ANOVA, $p_{\mathrm{BH}}=0.0431$) reach statistical significance. Overall, partial eta squared values were consistently above the conventional threshold for a large effect (0.14) for most muscles, indicating substantial effect sizes, even in cases where statistical significance was not reached, due to variability or sample size limitations.

Muscle activation was most effectively reduced by the force scaling intervention in the distal muscles, including the biceps, wrist flexors, and wrist extensors. In contrast, the posterior deltoid, trapezius, anterior deltoid, and triceps showed smaller or non-significant reductions, reflecting their primary role in supporting and stabilizing the shoulder and arm during task execution rather than directly controlling probe movements.

### 5.2. Teleoperation-Related Stress Analysis

The stress values derived from the sum of responses to the seven-item Likert-scale questionnaire are shown in Figure 9, computed for each subject in both non-scaled (in red) and scaled (in blue) conditions, as described in Section 4. The graph clearly shows that all subjects reported higher stress scores in the non-scaled condition compared to the scaled one. This result indicates a higher perceived stress level during task execution without force scaling, suggesting that the application of scaling helps reduce the cognitive and physical load experienced by the user in a teleoperation task.

To compare stress levels between the non-scaled and scaled conditions for each individual questionnaire item according to Section 4, a Wilcoxon signed-rank test was performed, whose result is shown in Figure 10. The graph displays asterisks indicating statistically significant differences between the two conditions: non-scaled (in red) and scaled (in blue). Corresponding unadjusted, Holm, and BH p-values for each item are reported in Table 3, where significance levels are denoted as follows: *** for *p* < 0.001, ** for *p* < 0.01, * for *p* < 0.05, and no asterisk for *p* ≥0.05. Additionally, effect sizes (r) were calculated to provide a standardized measure of the magnitude of the observed differences. According to common benchmarks, r values of approximately 0.1, 0.3, and 0.5 represent small, medium, and large effects, respectively, allowing a clearer interpretation beyond statistical significance alone, as shown in Table 3. Starting from the Holm correction,, statistically significant differences were found for questions 2, 3, 5, 6, and 7, indicating a clear reduction in perceived stress in the scaled trials compared to the non-scaled ones. Effect size r values for these items were large (above 0.5), confirming that the observed differences are not only statistically significant but also practically meaningful. In fact, all those items relate to worry, task execution difficulties (exceeding the operator’s possibilities or the operator’s control), and mental stress, which are all factors that directly influence the task-dependent perceived stress. Thus, a significant lower score for those items showed a clear teleoperation-related stress reduction. In contrast, no statistically significant differences were observed for questions 1 and 4, which addressed the sensation of pressure during the task and feelings of agitation or nervousness during the trial, respectively. The same considerations can be made using the Benjamini–Hochberg correction, although it yielded slightly lower p-values compared to the Holm correction. However, effect size r values for these items were medium to large (around 0.5), suggesting a potentially meaningful difference that might not have reached significance due to sample size or variability. This is probably due to the fact that a laboratory reconstruction of the task, however realistic, does not make the task seem completely real.

The results of the Wilcoxon signed-rank test applied to compare the total stress scores for all subjects in the two conditions are shown in Figure 11. This plot shows that, overall, there is a high statistical significance in the difference between the questionnaire responses for the non-scaled (in red) and scaled (in blue) conditions, with a reported *p*-value of 0.0009. In fact, performing the task in the non-scaled condition induces a significantly higher level of stress compared to the scaled condition. Furthermore, internal consistency was assessed using Cronbach’s alpha, which indicated acceptable reliability for both conditions: α = 0.86 for the non-scaled condition, and α = 0.59 for the scaled condition. These results support the overall reliability of the adapted seven-item scale across both experimental conditions.

## 6. Discussion and Conclusions

This study investigated the effect of applying a force scaling factor to the haptic feedback in a tele-ultrasound system, with the goal of reducing muscle fatigue and task-related stress during remote abdominal US procedures. In fact, ergonomics is a well-established issue in the ultrasound field [49]. This was already considered in previous tele-ultrasound systems, even if not considering the presence of force feedback to the user [50]. While force scaling inherently reduces the transparency of the teleoperation architecture (a drawback in most telemanipulation contexts), the experimental findings clearly demonstrate that this trade-off results in significant benefits. In particular, the reduction in both muscular fatigue and perceived stress was statistically evident across participants. This highlights an important opportunity: although tele-ultrasound has been extensively studied over the past two decades, most efforts have focused on accurate remote task replication and patient safety (both critical priorities) while rarely leveraging robotic technologies to improve the working conditions of the medical operator. In this work, particular emphasis was placed on that often overlooked aspect. The authors also conducted a meticulous operational and biomechanical analysis of the abdominal US procedure, which allowed for a faithful reconstruction of the task in the laboratory.

The results confirmed the effectiveness of the proposed approach. According to the Holm correction, a a statistically significant reduction in muscle activation was observed in 3 out of 7 monitored muscles during task execution. These trials involved 11 participants instructed to replicate, with high fidelity, a standard abdominal US procedure. In parallel, perceived stress during the task was also significantly reduced under the scaled condition. These two effects are likely interconnected: lower psychological stress may reduce the tendency to co-contract or stiffen the upper limb muscles, thereby lowering overall muscular effort. Improving the operator’s comfort in terms of both physical fatigue and stress is critical for ensuring high-quality diagnostic services. In fact, reduced muscular strain facilitates more consistent and fine-tuned contact force regulation, which directly impacts the quality of the US images and, by extension, diagnostic accuracy. Additionally, decreasing musculoskeletal load may help mitigate the long-term risk of work-related musculoskeletal disorders (WRMSDs), which are highly prevalent among sonographers due to the physically demanding nature of the task.

From a teleoperation-related stress point of view, the analysis of the questionnaire results showed that the scaled condition was associated with lower perceived stress scores for most items, particularly those related to task difficulty, mental workload, and worry. These findings suggest that the scaled condition may help manage task-dependent and cognitive stress during teleoperation, highlighting the potential benefits of scaling mechanisms in remote ultrasound activities. Although Cronbach’s alpha indicated slightly lower internal consistency for the scaled condition (α=0.59), this estimate reflects the characteristics of this study, which was conducted with a limited number of participants. Nevertheless, the overall pattern of results, combined with the adaptation of all items from psychometrically validated instruments, indicates that the questionnaire provided reliable insights into participants’ perceived stress across both experimental conditions.

In this study, the biomechanical analysis supported experiments focused on the abdominal region, widely recognized as the most demanding anatomical district due to the presence of air, fecal matter, and adipose tissue that impede ultrasound wave transmission, typically requiring higher contact forces compared to other procedures [46]. However, the proposed approach can be readily extended to other anatomical regions. Examinations of the neck, musculoskeletal system, or heart, for example, generally require lower forces, as the acoustic window is provided by superficial structures or bony interfaces [37]. Although different types of probes may be used, the fundamental principle of contact force modulation remains unchanged. Therefore, the benefits of force scaling in reducing muscular fatigue and task-related stress are likely to apply to other ultrasound examinations as well, highlighting the broader relevance of the methodology.

Operatively, the adoption of a tele-ultrasound system with a force scaling factor allows for the improvement of sonographers’ well-being and quality of life, contributing to the long-term prevention of WRMSDs. Compared to both conventional US examinations and remote procedures performed without force scaling, our results show that the application of the scaling factor significantly reduces muscle effort and perceived stress. However, it is fundamental to outline how the proposed force scaling method can be used effectively. The scaling factor can be freely adjusted by the physician, who may customize the level of force feedback according to personal needs related to workload management and daily fatigue, as well as to the specific anatomical district and body type of the patient under examination, which can vary significantly. This scaling should be considered by the user as a dynamic parameter, allowing for US procedure-dependent and patients’ physiological properties-dependent tuning to obtain the most advantageous ratio between transparency and perceived fatigue. Also, this parameter is modifiable during the sonographer’s workday according to the perceived fatigue. An initial guess of the parameter may be given after a familiarization phase, and its integration in the setup should be as easy as possible (e.g., control of the parameter through a simple slider). This enables the operator to perform examinations with more natural and sustainable postures over time. The system allows for the management of consistent and prolonged workloads, reducing the risk of injuries associated with incorrect postures or excessive force application, while supporting the performance of long shifts without compromising clinical outcomes. The integration of force scaling within a haptic tele-ultrasound interface can also facilitate the delivery of diagnostic services in remote or rural areas. By reducing physical effort and perceived workload, this approach enhances the feasibility of remote examinations, improving access to US services and ensuring the long-term sustainability of the operator.

Even if the findings are promising, allowing for a concrete and effective strategy to handle fatigue and stress-related issues in ultrasound remote procedures, two critical points emerged from the analysis that require further investigation.

First, transparency was deliberately sacrificed in favor of operator well-being. While this may seem counterintuitive in teleoperation design, the choice is justified by the fact that the primary feedback in US procedures is the image itself, while force feedback stays fundamental, mainly contributing to the naturalness and intuitiveness of the task execution [42]. A scaled feedback signal (attenuated in magnitude but consistent in dynamic response) appears to offer an effective compromise, enabling the operator to regulate contact force intuitively and without excessive physical load. In this study, the force scaling factor was kept constant across participants to control variability and accurately assess its effect on muscle fatigue and perceived stress. However, in practical clinical applications, the scaling factor can be adjusted by the sonographer during a preliminary phase of the ultrasound examination, allowing a flexible and personalized balance between maintaining sufficient feedback transparency and minimizing physical effort. Therefore, the visualization of ultrasound images remains unchanged; what is modulated is solely the force perceived by the sonographer. A controlled reduction in force feedback transparency can thus be tolerated to alleviate fatigue, providing ergonomic benefits without compromising the clinical execution of the task or the quality of the diagnosis. As supported in the literature [39], teleoperation performance depends not necessarily on maximum transparency, but rather on how effectively the feedback supports successful task completion.

Second, according to the Holm correction, four muscles did not show statistically significant changes (posterior deltoid $p_{\mathrm{Holm}}=0.1322$, trapezius $p_{\mathrm{Holm}}=0.1338$, anterior deltoid $p_{\mathrm{Holm}}=0.0923$, and triceps $p_{\mathrm{Holm}}=0.1033$). This result reflects the role of the shoulder in the motor control of the upper limb. Regardless of the testing conditions (scaled or non-scaled), the shoulder complex shows, on average, higher activation levels compared to the other muscles analyzed, as shown in Table 2, due to its dual function of limb positioning in space and joint stabilization. Repetitive and prolonged movements involving the distal muscles (forearm, wrist, and hand) lead to localized fatigue, which triggers compensatory strategies by the proximal muscles, particularly those of the shoulder and trunk [51]. This compensation consists of a process in which the motor control system redistributes the load by recruiting alternative muscle groups or expanding existing activation patterns to preserve task precision and joint stability [52]. Although this behavior of the shoulder muscles was expected, a different pattern was observed for the triceps. In fact, considering the Benjamini–Hochberg correction, it was observed that even though the shoulder shows high activation in both conditions (scaled and non-scaled), as shown in Table 2, the anterior deltoid, which is part of the same muscle complex, exhibits a significant difference ($p_{\mathrm{BH}}=0.0431$). The same can be observed for the triceps ($p_{\mathrm{BH}}=0.0431$). This behavior may be related to the fact that muscles influence each other during movement, so that after controlling for false positives, the activation of some muscles is not statistically separable from that of others. Moreover, the high effect size values observed for all the muscles analyzed, including the posterior deltoid ($\eta_{p}^{2}$ = 0.299), trapezius ($\eta_{p}^{2}$ = 0.210), anterior deltoid ($\eta_{p}^{2}$ = 0.387), and triceps ($\eta_{p}^{2}$ = 0.406), suggest that, with a larger number of participants, these muscles might also have reached statistical significance. Consequently, these findings indicate that the reduction in fatigue induced by the tele-ultrasound system under the scaled condition is not limited to distal muscles, but may also extend to the proximal musculature of the upper limb, suggesting that the system reduces fatigue along the entire kinematic chain of the arm. Specifically, biceps also show significant reductions in activation (biceps $p_{\mathrm{Holm}}=0.0469$), indicating that the scaled condition alleviates the effort required to stabilize the elbow and control forearm movements. From a biomechanical perspective, it should also be noted that, although some distal muscles, such as the wrist flexors and extensors, show high activation levels, as can be seen in Table 2, this contraction is primarily related to their role in fine and precise movements necessary to maintain contact with the probe and apply the desired force. In general, it is plausible that fatigue in the shoulder muscles is further exacerbated by the need to support the weight of the arm throughout the task, imposing a postural load in addition to the task-specific effort, as can be seen also in Figure 7. The use of a tele-ultrasound architecture, compared to performing conventional US examinations, allows the operator to adopt a more ergonomic and relaxed posture, thanks to the use of the scaling factor. In US practice, this condition is amplified by the fixed positioning of the monitor, which forces the operator to maintain physiologically demanding postures for prolonged periods.

The proposed study suffers from practical limitations mostly related to the number of involved subjects and their non-medical background and to the scenario conditions (no delay was considered, and the tests were conducted over a phantom).

The sample size of the participants and their background may raise some limitations. The number of participants nevertheless allowed for significant results, although it may have constrained the observation of potential additional effects. The high effect size observed suggests that increasing the sample size could enhance the significance of the results and more robustly confirm the effects detected, both in terms of muscle fatigue and perceived stress. To further assess the robustness of the results, the statistical power was calculated for each monitored muscle and indicated that only three muscles (anterior deltoid, wrist flexors, and wrist extensors) reached a statistical power above the commonly accepted threshold of 0.8, suggesting a high probability of detecting a true effect in these muscles. Conversely, the remaining four muscles exhibited lower power. Therefore, although the results for some muscles are robust, increasing the number of participants in future studies would likely improve the statistical power for the other muscles, allowing a more comprehensive evaluation of the effects of force scaling across the entire kinematic chain of the upper limb. Another aspect concerns the fact that this study involved participants with no prior experience in US. While this choice helped control variability, we acknowledge that experienced sonographers may adopt different postural habits, force application patterns, and ergonomic strategies. This represents a limitation in terms of ecological validity. Also, in the present study, US image visualization was deliberately excluded to avoid cognitive interference with the primary task of maintaining the prescribed contact force, which was continuously displayed in real-time.

Also, realistic conditions in terms of patient and delay presence deserve to be further investigated. In fact, experiments using real patients instead of phantoms will open opportunities to explore psychological and ergonomic implications not only for the operator but also for the examinee. Moreover, realistic tele-ultrasound tests should be conducted under conditions that closely replicate clinical practice, with the system being used remotely and subject to communication delays typical of tele-medicine applications. Operating the tele-ultrasound system under realistic conditions would allow a comprehensive evaluation of the effects of force scaling on operator muscle load and stress, providing a full assessment of the aims of the present study.

Those limitations, even if present, do not reduce the concreteness of the results found. On the contrary, these limitations well define the future development direction. In fact, future studies will actively involve professional sonographers to validate the findings in real clinical settings. This will also allow the investigation of probe handling techniques, image quality assessment, and operator comfort under authentic working conditions, thereby enhancing the translational impact of this research. In future experiments with professional sonographers, real-time US image visualization will be included, enabling the simultaneous assessment of operator fatigue, image quality, and diagnostic performance. Also, the experiments in future studies will be performed on real patients and in a delayed condition in a more realistic scenario, requiring the implementation of passivity layers [53,54] and facing concrete implementation issues [55].

In conclusion, this study demonstrates that the implementation of a force scaling factor in a bilateral tele-ultrasound system effectively reduces muscle fatigue and perceived stress, contributing to the long-term prevention of WRMSDs and promoting more natural and sustainable operator postures. The findings, even if affected by practical and task-modeling-dependent limitations, show a good impact on the analyzed problem and have practical implications for improving sonographer well-being, enabling safer prolonged shifts, and facilitating the delivery of remote diagnostic services in underserved areas.

## Figures and Tables

**Figure 1 sensors-25-05894-f001:**
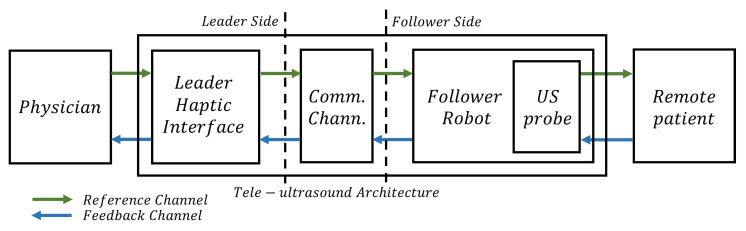
Block representation of a tele-ultrasound architecture. The physician directly interacts with the haptic interface, which sends a reference over the communication channel. The follower remote robot interacts through the US probe with the patient according to the reference received. The result of this interaction (typically a force recorded by a sensor) is sent back as feedback to the physician along with the US images.

**Figure 2 sensors-25-05894-f002:**
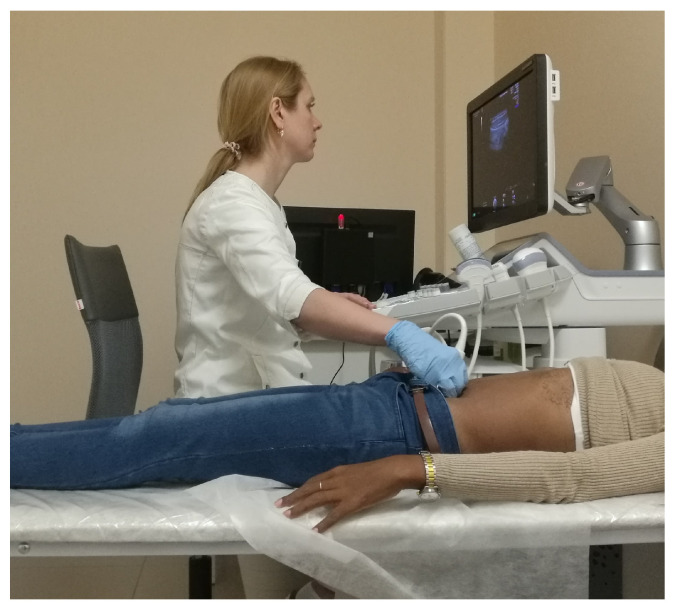
A doctor performs an abdominal US exam on a patient (Image credit: RG72 Eugeniy Roshkov via Wikimedia Commons—CC BY-SA 4.0). The adopted posture illustrates the typical static upper limb configuration used throughout the procedure.

**Figure 3 sensors-25-05894-f003:**
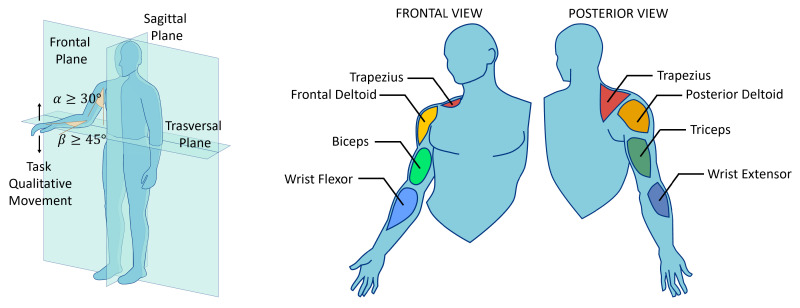
On the (**left**): typical starting posture of the physician during an abdominal US procedure, characterized by a 90° elbow flexion angle, shoulder abduction α greater than 30°, and forearm external rotation β exceeding 45°; on the (**right**): main muscles involved in the movement, namely trapezius, anterior and posterior deltoid, wrist flexors, wrist and extensors.

**Figure 4 sensors-25-05894-f004:**
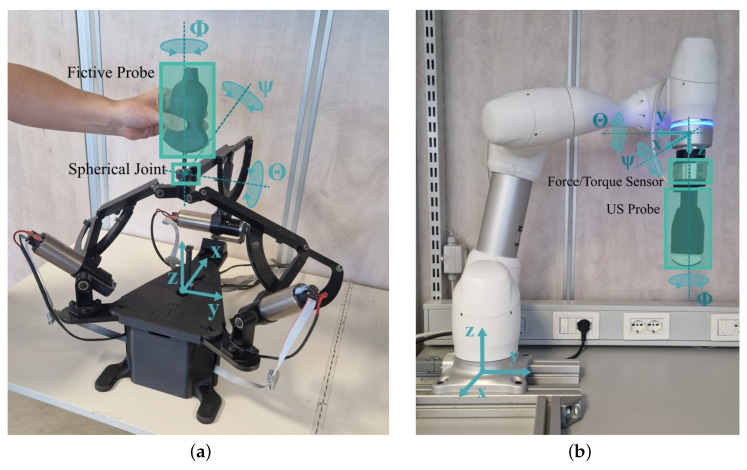
Custom tele-ultrasound architecture. The system allows the sonographer complete pose control and renders back interaction forces recorded through the sensor. (**a**) Leader haptic interface. A fictive probe has been installed through a spherical joint to allow the user complete, realistic control of the remote US probe. (**b**) A follower robotic arm equipped with a force/torque sensor and a US probe at the end-effector. The arm replicates the leader’s pose and measures force interactions with the phantom.

**Figure 5 sensors-25-05894-f005:**
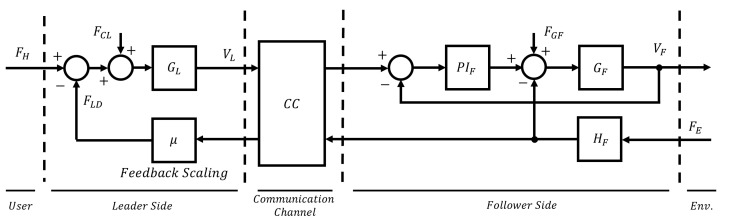
Block diagram representation of the teleoperation architecture. The feedback scaling factor μ is a constant value selected to reduce the force measured by the sensor and sent back as feedback. Thus, even if transparency (the quality of the force feedback) is partially reduced, the sonographer is still informed about remote interaction with the phantom.

**Figure 6 sensors-25-05894-f006:**
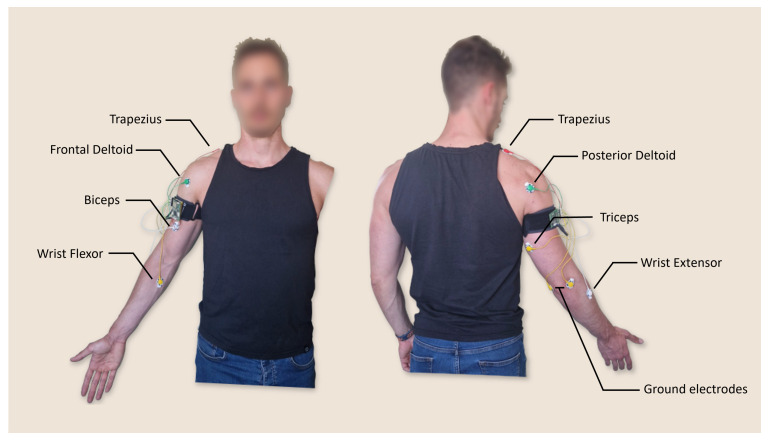
On the (**left**): frontal view of the subject with electrodes positioned on the trapezius, anterior deltoid, biceps, and wrist flexor muscles; on the (**right**): posterior view with electrodes for the trapezius, posterior deltoid, triceps, and wrist extensor muscles, including the two ground electrodes on the epicondyles.

**Figure 7 sensors-25-05894-f007:**
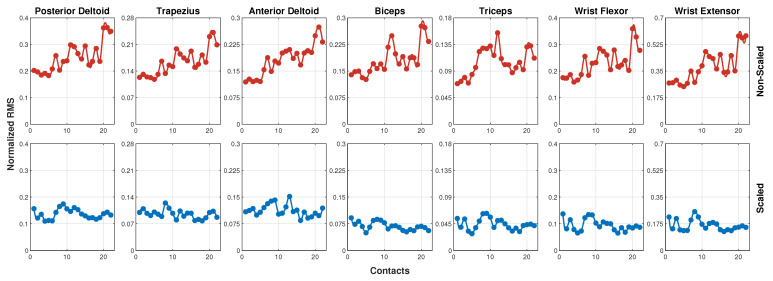
Normalized RMS values of muscle activation for each contact, trial, and muscle, shown for subject 2 in the two configurations: non-scaled (**top in red**) and scaled (**bottom in blue**). The plots preserve vertically the order of magnitude of the non-scaled condition, being the greatest. In addition to the magnitude difference, which is evident, the first row points out the fatigue effect in all muscles. This is evident from the increasing RMS values over contacts. Conversely, this effect is not evident in the second row since the scaling factor reduces fatigue over time.

**Figure 8 sensors-25-05894-f008:**
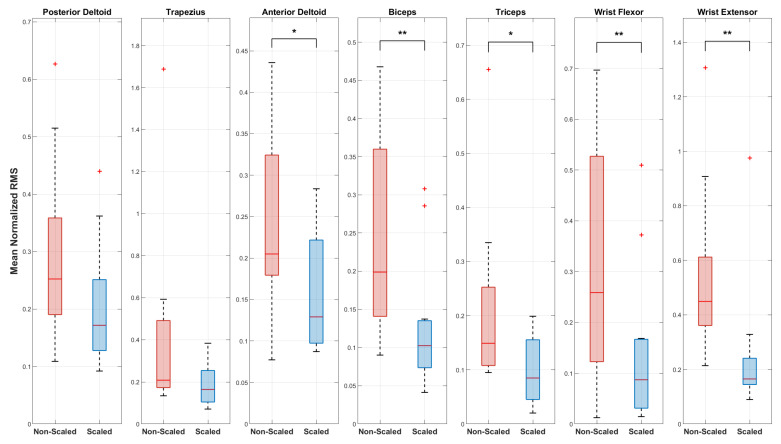
Distribution of normalized mean RMS values for each muscle, shown as boxplots for the two experimental conditions: non-scaled (**in red**) and scaled (**in blue**). Data refer to the statistical analysis performed on all subjects using RM ANOVA, before any correction. Significance levels for *p*-values are indicated as follows: ** *p* < 0.01, * *p* < 0.05, and no asterisk for *p* ≥ 0.05.

**Figure 9 sensors-25-05894-f009:**
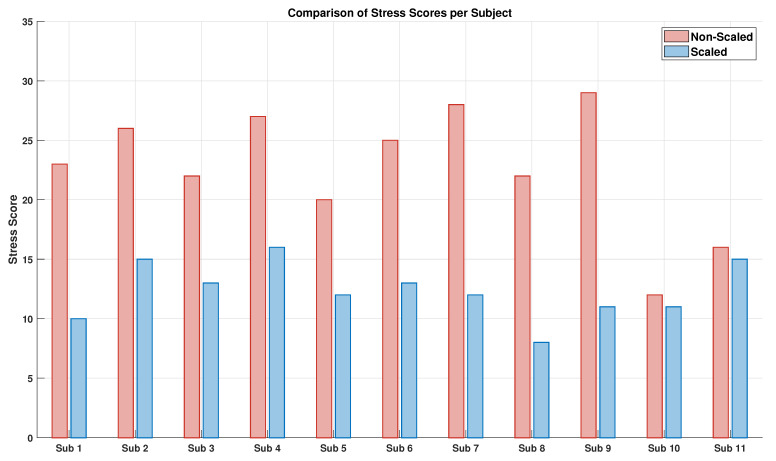
Stress scores per subject, obtained by summing the responses to the 7-item Likert-scale questionnaire, for both the non-scaled (**in red**) and scaled (**in blue**) conditions.

**Figure 10 sensors-25-05894-f010:**
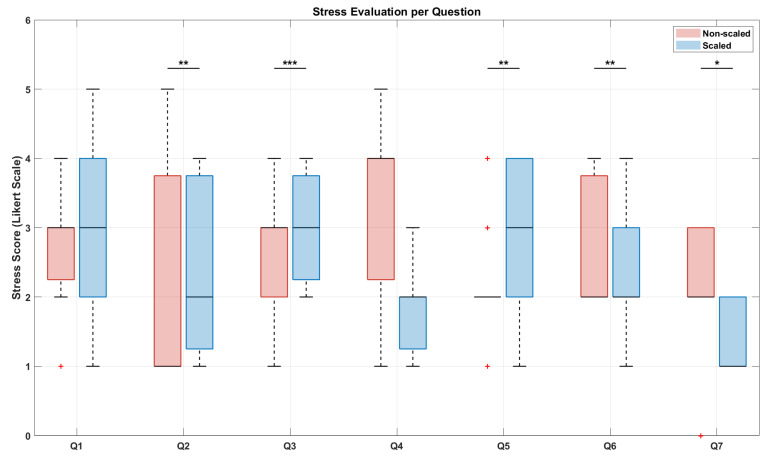
Wilcoxon signed-rank test comparing stress levels between non-scaled (**in red**) and scaled (**in blue**) conditions for each individual questionnaire item for all the subjects. Significance levels for *p*-values are indicated as follows: *** *p* < 0.001, ** *p* < 0.01, * *p* < 0.05, and no asterisk for *p* ≥ 0.05.

**Figure 11 sensors-25-05894-f011:**
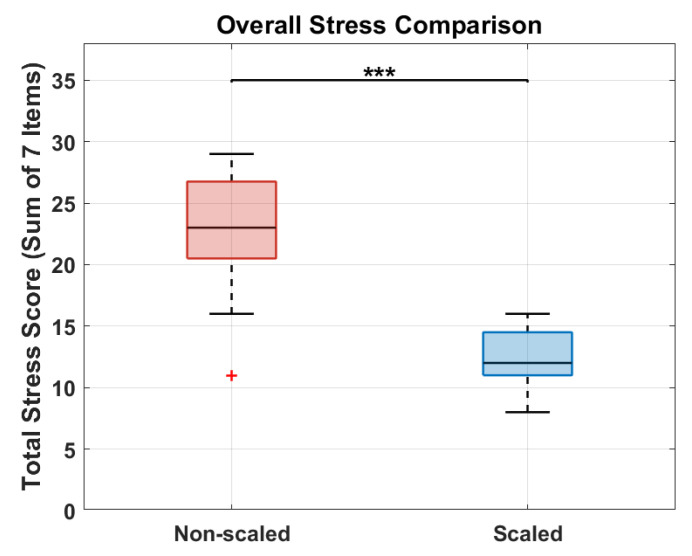
Wilcoxon signed-rank test comparing stress levels between non-scaled (**left in red**) and scaled (**right in blue**) conditions. Significance levels for *p*-values is indicated as follows: *** *p* < 0.001.

**Table 1 sensors-25-05894-t001:** Self-report questionnaire items and their source instruments.

Item	Adapted Statement for Teleoperation-Related Stress	Source
1	I felt under pressure during the task.	PSS / STAI
2	I was worried about making mistakes.	STAI
3	I felt overwhelmed by the difficulty of the task.	PSS
4	I felt nervous or agitated while operating.	STAI
5	It seemed like the demands exceeded my abilities.	PSS
6	I felt frustrated by the lack of control.	NASA-TLX Frustration
7	I experienced significant mental stress during the task.	PSS / NASA-TLX

**Table 2 sensors-25-05894-t002:** EMG results for each muscle. The mean ± standard deviation (SD) of muscle activation (%MVC) is reported for the scaled and non-scaled conditions. Statistical analysis was performed using Repeated Measures ANOVA (RM ANOVA). Reported unadjusted, Holm, and Benjamini–Hochberg (BH) *p*-values correspond to each comparison, with significance levels indicated by asterisks: ** for p<0.01, * for p<0.05, and no asterisk for p≥0.05. Partial eta squared ($\eta_{p}^{2}$) values are provided as measures of effect size, with conventional benchmarks of approximately 0.01 (small), 0.06 (medium), and 0.14 (large) effects.

Muscle	Mean ± SD Non-Scaled	Mean ± SD Scaled	Unadjusted *p*-Value	Holm *p*-Value	BH *p*-Value	$\eta_{p}^{2}$
Posterior Deltoid	32.1 ± 14.7	17.7 ± 11.4	0.0661	0.1322	0.0771	0.299
Trapezius	29.6 ± 15.4	16.0 ± 8.7	0.1338	0.1338	0.1338	0.210
Anterior Deltoid	26.9 ± 18.2	12.5 ± 4.9	0.0308 *	0.0923	0.0431 *	0.387
Biceps	26.7 ± 14.7	12.3 ± 8.0	0.0078 **	0.0469 *	0.0201 *	0.524
Triceps	26.6 ± 32.3	9.1 ± 6.9	0.0258 *	0.1033	0.0431 *	0.406
Wrist Flexor	34.7 ± 25.1	12.8 ± 15.0	0.0086 **	0.0430 *	0.0201 *	0.515
Wrist Extensor	57.7 ± 34.7	24.0 ± 28.0	0.0065 **	0.0453 *	0.0201 *	0.540

**Table 3 sensors-25-05894-t003:** Wilcoxon signed-rank test *p*-values (unadjusted, Holm, and BH) and effect size *r* per questionnaire item. Significance levels for *p*-values are indicated as follows: *** p<0.001, ** p<0.01, * p<0.05, and no asterisk for p≥0.05. Effect size (*r*) values of approximately 0.1, 0.3, and 0.5 correspond to small, medium, and large effects, respectively.

Item	Unadjusted *p*-Value	Holm *p*-Value	BH *p*-Value	Effect Size *r*
1 (Sensation of pressure)	0.0703	0.0703	0.0703	0.546
2 (Fear of making mistakes)	0.0020 **	0.0120 *	0.0068 **	0.932
3 (Feeling overwhelmed)	0.0010 ***	0.0070 **	0.0068 **	0.992
4 (Feeling agitated or nervous)	0.0625	0.1250	0.0703	0.562
5 (Task exceeds abilities)	0.0039 **	0.0195 *	0.0068 **	0.870
6 (Frustration from loss of control)	0.0039 **	0.0156 *	0.0068 **	0.870
7 (Mental stress experienced)	0.0156 *	0.0468 *	0.0218 *	0.729

## Data Availability

The raw data supporting the conclusions of this article will be made available by the authors upon request.

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
