# Peer review of "Impact of Force Scaling on Physician Fatigue in a Bilateral Tele-Ultrasound System"

_sensors, 2025, doi:10.3390/s25185894_

Round 1

Reviewer 1 Report

Comments and Suggestions for Authors

Review: The article titled “Force-Scaled Tele-Ultrasound: A Robotic Architecture for Reducing Operator Fatigue and Stress” presents a well-conceived and technically robust investigation into the application of robotic force-scaling feedback as a strategy to reduce musculoskeletal strain and perceived stress in tele-ultrasound procedures. The study is particularly relevant in the context of growing global interest in remote diagnostics and the urgent need to address the high prevalence of work-related musculoskeletal disorders (WRMSDs) among sonographers.

The manuscript demonstrates several strengths. The experimental design is thoughtfully structured, combining biomechanical task analysis, surface electromyography (sEMG), and subjective stress evaluation in a within-subject comparison between scaled and non-scaled force feedback conditions. The use of normalized EMG data and clear segmentation of task phases ensures methodological rigor. The findings are promising, with consistent evidence that the application of force scaling leads to a reduction in upper-body muscle activation and subjective stress levels, particularly in muscles associated with shoulder and upper-arm stabilization. These results provide meaningful insight into the ergonomic potential of robotic tele-ultrasound systems. Nonetheless, to enhance the clarity, scientific rigor, and overall impact of the manuscript, several aspects should be addressed. First, the Discussion and Conclusions section is notably brief and does not fully explore the implications of the findings. A more in-depth interpretation of the differential muscular responses, critical reflection on methodological limitations (e.g., absence of latency, use of novice participants, lack of diagnostic image evaluation), and integration of recent literature would significantly strengthen the scientific narrative. Furthermore, several interpretive elements currently placed in the Methods section—such as justifications for anatomical focus and procedural constraints—would be more appropriately developed in the Discussion.

Second, the presentation of results would benefit from greater statistical transparency. It is strongly recommended that the authors report descriptive statistics (mean ± standard deviation), exact p-values, and the specific statistical test used for each comparison, including non-significant results. Additionally, organizing these results into a clear summary table would facilitate interpretation and improve overall readability. Finally, although the manuscript is generally well written, a thorough revision by a native English speaker or professional editing service is encouraged to further refine style, grammar, and technical clarity. In summary, this study addresses a timely and important topic with sound methodology and meaningful results. With the incorporation of the suggested revisions, particularly in the Discussion and statistical reporting, the manuscript would offer a valuable contribution to the development of operator-centered, ergonomically optimized robotic systems for tele-ultrasound applications.

Below, I provide a structured set of comments and suggested revisions, organized according to each major section of the manuscript.

Introduction

In lines 24–25, the authors state: “These systems help extend access to diagnostic services not only for routine screening, but also in emergency scenarios…” The phrase could benefit from improved conciseness. Consider revising to: “These systems help extend access to diagnostic services for routine screening and in emergency scenarios…”

In lines 27–38, the authors provide a broad overview of the tele-ultrasound system architecture.
Although the description is technically sound, the section currently lacks updated citations beyond reference [1], which dates back to 2001. Given the rapid advancements in teleoperation systems and haptic feedback over the past two decades, it is recommended that the authors include more recent literature to support this section.

In lines 42–43, the authors state: “...making the procedure more natural, realistic, and precise despite the distance between leader and follower devices…” The term “precise” in this context requires clarification. Is the precision being referred to related to the spatial accuracy of the probe movement, the real-time synchronization of force feedback, or the diagnostic image quality? The authors are encouraged to specify the nature of this “precision” to avoid ambiguity. Additionally, the sentence lacks a period at the end and should be corrected for grammatical completeness.

In line 43, the authors state: “In traditional (local) US examinations, sonographers often suffer from work-related musculoskeletal disorders (WRMSDs).” It is suggested that a citation be added here specifically to support this statement, which is distinct from the more general overview that follows. For example, the reference by Evans et al. [5] or Zhang & Huang [6] may be appropriate to substantiate the prevalence of WRMSDs among sonographers.

In lines 60–62, the authors write: “...a sonographer performs between 10 and 20 US exams per day [16,17] during an 8-hour shift [17].” The use of references [16] and [17] in the middle of the sentence may disrupt readability. It is suggested to consolidate both citations at the end of the sentence. For example: “...a sonographer performs between 10 and 20 US exams per day during an 8-hour shift [16,17].”

In lines 65–72, the authors discuss the limitations of current tele-ultrasound systems in addressing WRMSDs. This paragraph presents several claims—such as the increased mental workload during teleoperation and the underutilization of robotics for ergonomic benefits—without supporting references. It is recommended to substantiate these statements with appropriate citations from the recent literature, for example, studies on cognitive load in teleoperation or ergonomic analyses of robotic medical systems.

In lines 65–67, the authors use the expression: “In fact…” twice in close succession.
To enhance stylistic quality and avoid redundancy, it is recommended to replace one of these occurrences with an alternative transition, such as “Indeed,”, “Notably,” or “It is well established that…”.

In lines 79–95, the authors elaborate on the potential of robotic force feedback to reduce operator fatigue and stress. While the content of this paragraph is highly relevant, its current form is dense and could benefit from improved structure and clarity. The paragraph attempts to cover the literature background, conceptual rationale, prior findings, and a gap in the state of the art all at once. It is suggested to restructure the paragraph by:

  1. First, summarizing the existing literature on force scaling in tele-ultrasound;
  2. Then, highlighting the lack of studies evaluating its impact on fatigue explicitly;
  3. Lastly, stating the motivation for the present study.

In lines 79–93, the authors state that the literature has proposed methods to address WRMSDs in tele-ultrasound, such as force scaling mechanisms, yet also mention that no studies have specifically evaluated the fatigue experienced by sonographers during such procedures. While these statements are not contradictory, the distinction between proposed ergonomic solutions and objective fatigue assessments may benefit from clarification. To enhance clarity and avoid possible confusion, the authors are encouraged to revise this passage as follows:

“Although the scientific literature presents tele-ultrasound approaches that aim to address WRMSDs, such as the use of force scaling in feedback mechanisms, there is, to the best of our knowledge, no study that directly and objectively assesses operator fatigue during the execution of tele-ultrasound examinations.”

Figures: I strongly recommend increasing the size of Figures.

Methods

General comment on the Materials and Methods section:

Across several parts of the Materials and Methods section—particularly in the introductory paragraph (lines 125–135), the subsection “Operational Aspects of Abdominal US Procedures” (lines 137–163), and portions of the biomechanical analysis (lines 164–197), the manuscript adopts a narrative and interpretive tone more typical of Introduction or Discussion sections. These passages provide valuable contextual background on the anatomical challenges of abdominal ultrasound, the rationale for selecting the region, and the nature of clinical procedures. However, they do not consistently convey the specific steps, parameters, and decisions involved in the experimental design of the present study.

In a Methods section, clarity and reproducibility are paramount. Therefore, it is essential to distinguish clearly between literature-based justification and actual methodological implementation. For instance:

  • When citing contact durations and applied forces from prior studies, it should be explicitly stated how these values informed the task reconstruction;
  • When referring to postural challenges or muscle involvement, the link between these observations and the selection of muscles for EMG recording should be made more procedural than theoretical.

The authors are encouraged to revise these portions to adopt a more objective and operational tone, explicitly detailing:

  • How the literature review was translated into experimental parameters;
  • What procedures were conducted (e.g., observations, measurements, protocols);
  • The sequence of steps followed to reconstruct the abdominal US task and evaluate biomechanical relevance.

Technical considerations regarding the Methods section (lines 125–233 and beyond):

While the methodological framework of the study appears technically sound and generally well-structured, a few aspects would benefit from additional clarification to enhance the transparency, reproducibility, and interpretability of the experimental design. The following points are offered as constructive suggestions, acknowledging that some decisions may have been made for practical or conceptual reasons that could be further explained in the final version:

  1. Absence of communication delay (line 289): The authors mention that no delay was introduced in the communication channel for the purpose of this study. While this approach may have been chosen to isolate the effects of force scaling, it should be noted that latency is a common and clinically relevant variable in real-world tele-ultrasound systems. The authors may wish to briefly acknowledge this limitation and clarify whether preliminary tests were conducted to estimate the potential impact of delay on performance or fatigue outcomes.
  2. Specification of the force-scaling factor μ (line 299): The use of a force-scaling factor is an important element of the experimental design. However, the exact value (or range) of μ applied during the experiment is not specified in the Methods section. Providing this information would improve methodological transparency and allow future studies to replicate the setup or compare results under similar conditions.
  3. Ultrasound image quality not assessed (line 322): The authors note that image interpretation was not required during the experimental task. While this is understandable in the context of an isolated ergonomic evaluation, it might be helpful to clarify whether any form of qualitative or technical validation was performed to ensure that the contact forces applied (with and without force scaling) still resulted in diagnostically acceptable image quality. This would help contextualize the trade-off between reduced operator load and clinical utility.
  4. Target force range (line 229): The adoption of a 6–12 N target range appears appropriate based on prior literature. Nonetheless, it would be helpful to elaborate on how participants were guided to maintain this range—e.g., whether visual force feedback was provided in real time, and if so, how this may have influenced task naturalness or cognitive load.
  5. Participant profile (line 364): The decision to recruit participants with no prior ultrasound experience is methodologically justified as a means of controlling for variability. However, given that experienced sonographers may adopt different postures, force patterns, or ergonomic strategies, the authors might consider noting this as a potential limitation in terms of validity. A brief mention that future studies could involve clinical professionals to validate these findings in real-world conditions would strengthen the discussion.

Overall, these points do not detract from the value or rigor of the study but are offered to enhance clarity and provide readers with a more complete understanding of the experimental parameters and their implications.

In line 147, the authors note: “...the average duration of contact phases with the patient is estimated to be between 2 and 5 min.” The source of this estimate is not clearly indicated. If this value is drawn from the literature, a reference should be added; if it is an observation made in the current study or a pilot phase, that should be clarified.

In line 229, the phrase reads: “...adopted a target force range between 6 and 12 N, depending on the body district and task.” Since this study focuses only on the abdominal region, the mention of “body district” may be unnecessary or misleading. Consider simplifying the sentence to: “...adopted a target force range of 6–12 N, based on values reported for abdominal ultrasound procedures.”

Statistical Analysis Section (lines 488–512):

The statistical approach adopted by the authors is overall appropriate and well-aligned with the study objectives. The use of Repeated Measures ANOVA (RM ANOVA) for the EMG data and the Wilcoxon signed-rank test for the subjective stress scores is methodologically justified and clearly reported. However, the following points may be considered to further improve the transparency and reproducibility of the analysis:

  1. Clarification of within-subject variance structure (RM ANOVA): While the manuscript states that a RM ANOVA was used to assess differences in muscle activation between scaled and non-scaled conditions, it would be helpful to specify whether sphericity assumptions were verified (e.g., Mauchly’s test), and whether any correction (such as Greenhouse–Geisser) was applied in the event of violation. Even if only two conditions are compared, an explicit note on this assumption strengthens the statistical rigor.
  2. Effect size reporting:The statistical results would benefit from the inclusion of effect sizes (e.g., partial eta squared or Cohen’s d) for the RM ANOVA and Wilcoxon test. This would provide the reader with a better understanding of the magnitude of the observed differences, beyond p-values alone.
  3. Correction for multiple comparisons: Given that multiple RM ANOVAs were performed (one per muscle), it would be advisable to indicate whether any adjustment for multiple comparisons was applied (e.g., Bonferroni or Holm correction), or to justify why it was not deemed necessary. The same applies to the item-level Wilcoxon tests.
  4. Justification for normality test (Kolmogorov–Smirnov): The use of the Kolmogorov–Smirnov (K–S) test to assess normality is acceptable but may be less sensitive than the Shapiro–Wilk test, especially for small samples. If the sample size was below 50, the authors may consider mentioning why K–S was preferred, or alternatively verifying results using Shapiro–Wilk as a robustness check.
  5. Internal consistency of the stress questionnaire: The authors note that Cronbach’s alpha was not computed due to the small sample and exploratory nature of the study. While this is understandable, even a preliminary estimation could provide valuable insight into the reliability of the adapted questionnaire. If not feasible, a brief mention of item–total correlations or an acknowledgment of this as a limitation may strengthen the psychometric transparency.

(These adjustments are not critical but are suggested to elevate the methodological transparency and statistical rigor of an otherwise well-conducted and meaningful analysis)

Results

In the subsection “Muscle Activation” (lines 513–537), the results are generally well presented and logically organized; however, several improvements are strongly recommended to enhance scientific transparency and interpretability:

  1. Include descriptive statistics (mean ± standard deviation) for all muscles and conditions:

Currently, the authors describe whether muscle activation increased or decreased, but do not report the actual mean and standard deviation values of EMG (%MVC) for the “scaled” and “non-scaled” conditions. This makes it difficult to assess the magnitude of the observed effects. For each muscle, the corresponding values (e.g., TRAP: 31.2 ± 8.4%MVC vs. 21.5 ± 6.7%MVC) should be explicitly stated.

  1. Report exact p-values and the statistical test used for each comparison: While it is noted that significant differences were observed for some muscles, the exact p-values are not provided, nor is the test type explicitly mentioned in the results text. Even though the methods section describes the use of repeated measures ANOVA, each reported comparison should specify the test applied and the resulting p-value. For example: “A significant reduction was found in TRAP activation (RM ANOVA, p = 0.012).”
  2. Include data for non-significant comparisons as well: Muscles such as EDC and ECU are described as having no significant change in activity, but no numerical values or p-values are presented. For completeness and consistency, the authors should report the corresponding descriptive statistics and exact p-values for these cases too. This is especially important to allow readers to assess whether non-significant results are due to small effect sizes or large variability.
  3. Standardize the reporting format for all muscles: The order in which muscles are presented appears arbitrary. Reorganizing them either by anatomical region (e.g., shoulder, arm, forearm) or by effect size (e.g., largest to smallest difference) may help improve readability and consistency. Additionally, using a uniform format for each result (mean ± SD, test type, p-value) would enhance clarity.
  4. Provide a summary table of EMG results: It is strongly recommended to include a table summarizing all EMG results per muscle, listing:
    • Muscle name
    • Mean ± SD for scaled and non-scaled conditions
    • Statistical test used
    • Exact p-value
    • Indication of statistical significance

This would offer readers a clear and accessible overview of the quantitative findings.

  1. Consider adding a brief interpretation at the end of the subsection:
    A short concluding sentence summarizing which anatomical regions (e.g., shoulder vs. forearm) were most affected by the force-scaling condition would provide a helpful transition into the subsequent subsection.

Discussion and Conclusions (lines 576–601):

While the overall structure of the study is technically sound and the results are promising, the Discussion and Conclusions section is notably underdeveloped. It currently offers only a superficial interpretation of the findings and does not fully leverage the methodological depth and experimental richness presented in earlier sections.

The following points are suggested to improve this section:

  1. Expand the interpretation of results: The current discussion briefly reports that muscle activation and subjective stress were reduced but does not explore in depth why specific muscles (e.g., shoulder vs. forearm) responded differently to force scaling. A more detailed biomechanical interpretation of these differential effects could provide valuable insight into load redistribution and fatigue mitigation strategies in tele-ultrasound systems.
  2. Critically address methodological limitations: Important methodological choices, such as the absence of simulated communication delay, the exclusion of trained sonographers, the lack of diagnostic image quality evaluation, and the small sample size, should be acknowledged and discussed. Even if these were justified for experimental control, reflecting on their implications helps situate the study within its appropriate scientific and clinical scope.
  3. Broaden the literature context: The discussion includes only three references, all of which were cited earlier in the manuscript. To enhance scholarly depth, the authors are encouraged to integrate additional recent studies on tele-ultrasound ergonomics, EMG-based workload assessments in remote healthcare, and robotic architectures involving force feedback. This would more clearly position the study within the current state of the field.
  4. Include practical and clinical implications: It would be valuable to discuss the translational potential of the findings. How might this robotic architecture improve sonographer well-being in actual clinical practice? Could it reduce occupational injury rates in high-volume settings or expand access in remote/rural environments? Highlighting these implications would elevate the real-world relevance of the study.
  5. Strengthen the conclusion with structured insights: The final paragraph should summarize the key contributions, acknowledge the main limitations, and propose concrete directions for future work—such as validating the system with professional sonographers, integrating real-time image analysis, or evaluating performance under simulated latency conditions.
  6. Balance conceptual content across sections: Notably, several interpretive and conceptual discussions are currently embedded in the Materials and Methods section, especially in the “Operational Aspects of Abdominal US Procedures” and “Biomechanical Analysis” subsections. These paragraphs contain valuable insights that would be better suited for the Discussion, where they can be critically interpreted in light of the experimental results. The authors are therefore encouraged to redistribute this content, ensuring that the Discussion becomes the main space for reflective analysis, while the Methods remain focused on procedural description.

Author Response

Dear Reviewer,

please find attached our replies to all the Reviewers. We hope we have answered all concerns satisfactorily.

Best regards,

The Authors

Reviewer 2 Report

Comments and Suggestions for Authors

General comments:

This manuscript explores the effect of force scaling on reducing physician fatigue and stress in a bilateral tele-ultrasound system. The authors performed comprehensive biomechanical and operational analyses, supported by experimental validation using surface electromyography (sEMG) and subjective stress assessments. The manuscript is clearly structured and includes effective figures to illustrate the findings. However, the innovative aspect of the research is moderate, as force-scaling techniques are already established in the teleoperation literature. Several points require further clarification to improve methodological transparency and practical applicability.

Major Comments:

  1. While the biomechanical analysis is well-done, the authors need to provide explicit justification for selecting specific muscle groups beyond general ergonomic considerations. Clarifying the relationship between the chosen muscles and specific movements involved in ultrasound examinations would enhance the methodological rigor.
  2. The rationale behind selecting the abdominal region for testing is briefly mentioned. It would be helpful to elaborate on how the results could extend to other ultrasound examinations or teleoperation scenarios, thereby increasing the study's broader relevance.
  3. The experimental design simplifies real clinical conditions significantly. The manuscript should clearly discuss limitations arising from these simplifications, particularly concerning actual patient variability and probe handling by experienced sonographers.
  4. Although force scaling effectively reduces fatigue, the discussion about reduced feedback transparency needs to be expanded. A thorough consideration of how decreased transparency might affect clinical decisions or overall task performance would improve the manuscript's depth.
  5. The study uses EMG and subjective questionnaires effectively, but the small sample size (n=11) may limit the robustness of the conclusions. The authors should discuss how the limited sample size impacts statistical power and the generalizability of their findings.
  6. Practical recommendations from the study are somewhat underdeveloped. The manuscript should clearly outline potential approaches for integrating the proposed scaling method into current tele-ultrasound practices or provide guidelines for clinicians.

Recommendation:

Major revisions are recommended. Addressing the comments above, particularly regarding methodological clarity, practical limitations, and practical implications, will substantially enhance the manuscript's quality and applicability.

Author Response

(The authors gave the same response as above.)

Round 2

Reviewer 1 Report

Comments and Suggestions for Authors

Dear Authors,

I have reviewed the revised submission. All issues raised in my prior review were addressed in a technical and professional manner. The manuscript is accepted for publication. Congratulations and thank you for the careful revisions.

Reviewer 2 Report

Comments and Suggestions for Authors

The author has provided detailed responses to the reviewers’ comments, and all of my questions have been clearly addressed. I believe the paper is now ready for publication.